# Personalized Self-Monitoring of Energy Balance through Integration in a Web-Application of Dietary, Anthropometric, and Physical Activity Data

**DOI:** 10.3390/jpm12040568

**Published:** 2022-04-02

**Authors:** Giada Bianchetti, Alessio Abeltino, Cassandra Serantoni, Federico Ardito, Daniele Malta, Marco De Spirito, Giuseppe Maulucci

**Affiliations:** 1Department of Neuroscience, Biophysics Sections, Università Cattolica del Sacro Cuore, Largo Francesco Vito, 1, 00168 Rome, Italy; giada.bianchetti@unicatt.it (G.B.); alessio.abeltino@unicatt.it (A.A.); cassandra.serantoni@unicatt.it (C.S.); marco.despirito@unicatt.it (M.D.S.); 2Fondazione Policlinico Universitario “A. Gemelli” IRCCS, 00168 Rome, Italy; 3RAN Innovation, Viale della Piramide Cestia, 1/c, 00153 Rome, Italy; c.f.ardito@gmail.com (F.A.); daniele.malta@raninnovation.com (D.M.)

**Keywords:** self-monitoring, energy balance, personalized medicine, web-application, weight loss, exercise, nutrition, metabolism, wearables, obesity

## Abstract

Self-monitoring of weight, diet and physical activity is a valuable component of behavioral weight loss treatment. The validation and user-friendliness of this approach is not optimal since users are selected from homogeneous pools and rely on different applications, increasing the burden and achieving partial, generic and/or unrelated information about their metabolic state. Moreover, studies establishing type, time, duration, and adherence criteria for self-monitoring are lacking. In this study, we developed a digital web-based application (ArmOnIA), which integrates dietary, anthropometric, and physical activity data and provides a personalized estimation of energy balance. Moreover, we determined type, time, duration, and adherence criteria for self-monitoring to achieve significant weight loss in a highly heterogeneous group. A single-arm, uncontrolled prospective study on self-monitored voluntary adults for 7 months was performed. Hierarchical clustering of adherence parameters yielded three behavioral approaches: high (HA), low (LA), and medium (MA) adherence. Average BMI decrease is statistically significant between LA and HA. Moreover, we defined thresholds for the minimum frequencies and duration of dietary and weight self-monitoring. This approach can provide the correct clues to empower citizens with scientific knowledge, augmenting their self-awareness with the aim of achieving long-lasting results when pursuing a healthy lifestyle.

## 1. Introduction

According to the latest WHO Global Health Observatory data, collated in 2016, more than 1.9 billion adults were overweight, with 650 million of these individuals being obese. The global prevalence of obesity nearly tripled between 1975 and 2016, with substantial rises in most countries, including those deemed to be low-income and middle-income nations. As such, the global spread of obesity has been labelled a pandemic, albeit one with a slower onset of cases and fewer detrimental effects than the 2009 H1N1 pandemic or the COVID-19 pandemic [1,2]. Evidence that relates to obesity is biased towards its causes rather than strategies for prevention, which have not yet been widely replicated or delivered at a scale that offers a clear option for public health strategies. Finding and implementing solutions necessitate multidisciplinary approaches stimulating effective behavior change, which is an important component of any response to obesity [3]. However, this is a complex process for individuals that goes beyond education and the provision of information. Achieving change is difficult, resource-intensive, and time-consuming. Interventions based on improved nutrition and increased physical activity can be effective for individuals if the provided information is personalized, and can be turned into effective actions, with an outcome that can be quantified and related to their own physiological status [4]. In this context, technology can help to support and maintain healthy behaviors: particularly smart wearable devices, can monitor and provide feedback on energy intake and energy expenditure [5]. For people who want to be healthier and to change their lifestyle, these devices will make it easier, as measurement and feedback systems are becoming more refined and personalized. These systems have the potential to be linked into a wide range of lifestyle-support services through community, public and private providers. Existing studies related to self-monitoring some of the, or all three, components analyzed in behavioral weight loss studies (diet, exercise, and self-weighing) reported a significant association between self-monitoring and weight loss [6,7,8,9,10]. However, the level of evidence in the existing studies is weak because methodological limitations are still present, particularly: (1) lack of personalization due to the scarce integration of the information provided to the user: users must rely on different applications, furnishing partial and unrelated information about their metabolic state, where energy intake and expenditure are not directly related; (2) the homogeneity of the cohort; (3) the lack of information on the adherence and duration criteria that must be met for the improvement to be effective. To overcome these limits, the objectives of this study are: (a) to develop a single digital application (ArmOnIA, www.apparmonia.com (accessed on 30 March 2022)) to collect, integrate and visualize anthropometric, nutritional, and physical activity data in a comprehensive way. Data flows from wearables (smart band and impedance balance) and diet diaries are collected and aggregated to build an accurate and personalized estimation of energy balance (accounting for individual body composition, age, and hydration state); (b) to determine the type, time, duration, and adherence criteria of self-monitoring with ArmOnIA in a highly heterogeneous group to achieve significant weight loss.

## 2. Materials and Methods

### 2.1. Study Population and Protocol

In this single-arm, uncontrolled prospective study, a group of 35 voluntary normal or overweight adults expressing the will to lose weight for several reasons was required to self-monitor their weight, diet, and step count each day for 210 days using the ArmOnIA app, without pre-determined objectives or intervention. For inclusion the study, the participants needed a starting Body Mass Index (BMI) ≥ 19 kg/m^2^. For each participant, the effective duration of the study was then determined as the number of days between the first and the last day of weight- and food-monitoring, and the weight frequency monitoring (WFM) and food frequency monitoring (FFM) were calculated as the number of days in which weight- and/or food-tracking occurred over the effective duration of the study.

All the participants were required to measure their weight every morning before breakfast, on an empty stomach, barefoot, and with the head not reclined for a few seconds. Contextually, they were asked to record their daily diet in as much detail as possible, including daily drinks, such as water and coffee.

### 2.2. Wearables and Devices

The following devices were chosen to track anthropometric and activities data:

MiBand 6, a smartband of Xiaomi^®^ industry (Beijing, China), for the tracking of heartbeat, through a photoplethysmography (PPG) integrated biosensor; blood oxygen saturation level; sleep stages (REM, deep, shallow) using heartbeat variations; calories burned through exercises (walking, running, etc.). The user must always keep the smart band on, as was explained in the protocol.

Mi Body Composition Scale, an impedance balance of Xiaomi^®^ industry (Beijing, China), was used to track anthropometric data, such as weight, resting metabolism, fat rate, muscle rate, bone mass, bmi.

These devices have already been used in three studies on PubMed, and 11 clinical trials were performed using MiBand [5]. Validation results for the estimation of RMR can be retrieved in a recent publication [6].

### 2.3. Web-App Development and Estimation of Personalized Energy Balance (EB)

A web application (www.apparmonia.com (accessed on 30 March 2022)) was developed in Python 3.8 with the libraries Django and Django-plotly-dash for data collection, storage, and the visualization of energy balance through a dashboard.

The web application allows for data collection, storage, analysis, and visualization. The details of each step are reported in the following.

(a)Data Collection
Data from wearables were fetched through the ZEPP API^®^.Food and other activities not included in the Smartband (home activities, music playing, driving, etc.) were provided in-person through a digital diary.(b)Data StorageFetched data underwent anonymization and storage into a NoSQL database (MongoDB^®^).(c)Data Analysis and Visualization

Figure 1 shows the food diary dashboard, where the user can keep track of their daily Energy Balance (EB), calculated according to the formula:EB = EI − TEE,(1)
where EI is the daily energy intake, and TEE is the daily total energy expenditure.

EI is considered the sum of all the ingested calories, as retrieved from the following databases: DIETABIT (www.dietabit.it (accessed on 30 March 2022)), CREA (www.crea.gov.it (accessed on 30 March 2022)), BDA (www.bda-ieo.it (accessed on 30 March 2022)), and OPENFOODFACTS (www.it.openfoodfacts.org (accessed on 30 March 2022)).

TEE is calculated according to the formula:TEE = RMR + TEA + TEF,(2)
where TEA is the Thermic Effect of Activity, and RMR is the Resting Metabolism Rate. These two terms were measured using the values provided by Huami (Xiaomi^®^ industry, Beijing, China) [6], while Thermic Effect of Food (TEF) refers to the energy expenditure related to food consumption (i.e., digestion, absorption, assimilation, and storage), depending on both the amount and the type of food consumed. This term, which accounts for about 10% of TEE, is estimated from food data through the following formula [7]:TEF = 0.95 × (*m*_C_ × 3.75) + 0.015 × (*m*_L_ × 9) + 0.25 × (*m*_P_ × 4),(3)
where *m*_C_ is the mass expressed in grams of total daily carbohydrate content, *m*_L_ is the mass expressed in grams of total daily lipid content, and *m*_P_ is the mass expressed in grams of total daily protein content.

In Figure 1A, the dashboard, realized through the Django-plotly-dash library, displays what the users see to be aware of their daily energy balance. The green bar represents the estimated RMR, while the yellow bar shows the additional amount of energy, estimated by either TEF or TEA, and deriving from the activity introduced by user (gardening, playing music, etc.). The energy intake is represented by the overlapping red bar. Through this scheme, the user is aware that, to lose weight, the TEE must be maintained under the vertical blue bar, representing the EE value. All these quantities are updated in real time. In Figure 1B, two-sample time series of the collected energy balance and weight records versus time are reported. From these curves, it is possible to observe the relationship that exists between the EB and the weight maintenance, gain, or loss. Indeed, if the EB is kept around zero, as between month 1 and 2 (M1–M2, green line, Figure 1B, above), no changes are observed in the average weight (Figure 1B, below). However, when the average EB is above zero for a prolonged period (M2–M3), weight gain occurs. On the contrary, keeping the EB below the zero level ensures significant weight loss, as can be observed between M3 and M6.

### 2.4. Statistics

Unsupervised clustering analysis and statistical tests among the three clusters were performed with Orange 3.26 (https://orangedatamining.com/ (accessed on 30 March 2022)) and R (https://www.r-project.org/ (accessed on 30 March 2022)). Differences between clusters were determined by conducting a Kruskal–Wallis test for the data that were not normally distributed. Normal data distribution was assessed by visual inspection, variance comparison and Shapiro–Wilk’s test. Subsequently, a Pairwise Wilcox Test was performed for post-hoc comparisons, and, for normally distributed variables, a one-way ANOVA with Tukey post-hoc analysis was conducted. The effect of self-monitoring on BMI variations in time was assessed by a repeated-measures ANOVA, which is the equivalent of the one-way ANOVA but for related, not independent, groups, followed by a paired *t*-test. *p*-values below 0.05 were considered statistically significant.

## 3. Results

### 3.1. Overall Effect of Self-Monitoring on BMI Variation

In this single-arm, uncontrolled prospective study, we evaluated the effect of weight- and food-frequency self-monitoring, respectively, on a heterogeneous population constituting 35 healthy volunteers with a starting BMI ≥ 19 kg/m^2^. The whole study group is constituted by 23 females (66%) and 12 males (34%) between 20 and 75 years (mean value = 37 ± 12 years), with an average starting BMI = 24.6 ± 3.8 (min = 19.3, max = 36.9). Although the whole period under analysis lasted 210 days, corresponding to 7 months, we defined the effective duration of the study for each participant as the number of days between the first and the last day of weight- and food-monitoring, observing a mean overall duration of 160 ± 53 days, with WFM and FFM assessed on 66 ± 27% and 70 ± 24%, respectively, of the overall population. Through a paired-sample *t*-test, we examined the association between app-based self-monitoring and weight change, retrieving a significant BMI variation of −0.4 ± 1.1 (*p*-value = 0.05), thus indicating that, in the heterogenous population we analyzed, the frequencies of weight and food monitoring, respectively, without additional interventions, can determine a significant decrease in BMI during the considered period. To this aim, we further stratified the population according to the weight- and/or food-frequency monitoring, to observe the potential effects that may arise from different adherence rates.

### 3.2. Hierarchical Clustering Reveals Three Main Behaviors towards Digital Self-Monitoring

To investigate the effects of adherence behaviour, we stratified the population according to WFM and FFM values by using an unsupervised hierarchical clustering method. The idea of cluster analysis is to measure the distance between each pair of participants in terms of WFM and FFM, and then to group subjects which are close each other. More specifically, based on the distance map reported in Figure 2, the clustering algorithm identifies the closest observations (i.e., subjects with similar WFM and FFM), and iteratively merges them within the same cluster until all clusters were merged. The distance map allows for the distances between objects to be visualized using colored spots instead of numbers. In the map, the Euclidean distances between rows, corresponding to participants in the study, are represented in the weight- and food-frequency monitoring data, with smaller distances colored in blue and larger ones shown in yellow. The matrix is symmetric, and the diagonal is constituted by black spots, since no user is different from himself in terms of WFM and FFM. In this map, elements are arranged by clustering with ordered leaves, which maximizes the sum of similarities between adjacent elements.

The clustering was performed based on the Ward method, which has proven to outperform other hierarchical methods in producing homogeneous and interpretable clusters [11]. The net operation yields a hierarchical classification tree [12], as reported in Figure 3A.

The hierarchical clustering of WFM and FFM yielded three different behavioural approaches to self-monitoring: a first cluster, colored in light blue, with a high average rate of self-monitoring adherence (WFM = 0.83 ± 0.12; FFM = 0.81 ± 0.15), indicated as HA (high adherence), a second cluster, colored in red, with a FFM = 0.70 ± 0.16 and a lower WFM = 0.60 ± 0.09, named medium adherence (MA), and a third cluster, represented in greed, characterized by low adherence (LA) to self-monitoring, with values of WFM = 0.37 ± 0.24 and FFM = 0.28 ± 0.07, respectively.

### 3.3. Description of the Subgroups

The obtained clusters were included in a comparative analysis, reported in Table 1. Following the hierarchical clustering and post-hoc comparison, the three clusters present different WFM and FFM values, but matched age, sex, duration of the study, starting BMI, and average daily steps. The percentage of cluster population, which is 63% (22/35) for HA, 20% (7/35) for MA, and 17% (6/35) for LA, with a χ^2^ < 0.001, as represented in Figure 3C, overall indicates a good engagement of participants.

We did not observe a significant variation in physical activity, monitored through the number of steps, in the three clusters, although this parameter was under continuous monitoring, and did not require any user engagement. Interestingly, the average BMI decrease was the highest in the HA cluster (−0.6 ± 0.9), a statistically significant value (*p*-value = 0.007) with respect to the LA cluster (0.6 ± 0.7) in the considered study period (see Table 1 and Figure 4).

### 3.4. Determination of the Average Time and Optimal Adherence Required to Achieve a Significant Weight Loss

To evaluate the minimum average self-monitoring time that was necessary to achieve a significant weight loss, we selected a subgroup of 13 participants belonging to the HA cluster, who monitored themselves for the entire study period (210 days). For each participant, the mean BMI was evaluated every 15 days, and the average values of the whole subgroup are reported in Figure 5. Since we aimed to compare BMI across users and over time, based on repeated observations of the same variable, we used a repeated-measures ANOVA that accounts for related, not independent, variables. This test, yielding a *p*-value < 0.001, was first used to assess the statistical significance of BMI variations in time, and was then followed by a paired *t*-test to define the minimum time required to reach a significant weight loss. As represented in Figure 5, at least 75 days of high-rate self-monitoring were required to reach a significant BMI variation, with mean values decreasing from 25.1 ± 3.2 to 24.7 ± 3.4 (*p*-value = 0.023). Interestingly, the plot reveals that, in the heterogeneous population that we considered, the BMI constantly decreased after day 75, reaching a minimum after 150 days (24.2 ± 3.3, *p*-value = 0.0005). However, although a slight increase was further observed, the final average BMI (day 210) was 24.4 ± 3.4, a significantly lower value with respect to the starting BMI (day 1, *p*-value = 0.01).

Moreover, this study also established a framework to define the optimal adherence to achieve clinically significant weight loss in healthy population by increasing self-awareness of one’s own metabolic status. A lower limit for adherence frequencies can be conceived as µ-σ, where µ and σ are the average and standard deviation of the HA cluster, respectively. Using our evidence, a FFM of (74–19)% and a WFM of (68–23)%, are enough to achieve a significant BMI loss of 2%.

## 4. Discussion

Self-monitoring of weight and dietary intake is a valuable component of behavioral weight loss and the maintenance of a healthy weight. Greater dietary self-monitoring adherence was associated with weight loss, but several limitations can affect these tools: lack of integration, the definition of adherence to dietary and weight self-monitoring, type (diet, weight, or both) and the minimum time required for self-monitoring for successful outcomes. In this study, we developed a single digital application (ArmOnIA, www.apparmonia.com (accessed on 30 March 2022)) to collect, integrate and visualize anthropometric, nutritional, and physical activity data in a comprehensive way. Data flows from wearables (smart band and impedance balance) and diet diaries are collected and aggregated to build an accurate and personalized estimation of energy balance (accounting for individual body composition, age, and hydration state). Here, we considered a cohort of 35 healthy volunteers to analyze the effects of self-monitoring on a heterogenous population, including people of different ages and RMR, and characterized by a wide range of BMIs, from normal to obese. The analysis of the whole cohort revealed that, apparently, self-monitoring and knowledge of personalized energy balance has only a marginal effect on BMI variation. However, the unsupervised clustering of frequency monitoring allows three behavioral approaches to be distinguished, differing in the frequency and type of self-monitoring: HA, characterized by a high average rate of both weight and food self-monitoring; MA, with a higher average rate of FFM with respect to WFM, and LA cluster, showing a generally low adherence to self-monitoring. Interestingly, this investigation revealed that the highest percentage of the enrolled population belongs to the HA cluster, including 22 of the 35 participants. This result allows us to observe an optimal baseline engagement in volunteers reducing their BMI by self-monitoring through the ArmOnIA app, which also ensures the maintenance of a temporary engagement for an average of 5.3 ± 1.8 months for all users. The dynamic calculation of personal energy balance, allowing for the integration of energy intake and energy expenditure, can encourage the user to continue self-monitoring for long periods; indeed, in the huge ecosystem of health applications, limitations on specific aspects (nutrition, sport, health) may limit the catchment area to fewer categories. Moreover, the use of complete, validated, and open access databases may increase user engagement regarding food self-education by learning complete nutrition facts regarding their intake, to adhere to the standards accepted by the scientific community. Although more focused studies, comparing the effects of different self-monitoring strategies, are needed to confirm this point, we believe that this study constitutes the first indication that the integrated collection of dietary, physical activity, and anthropometric quantities is necessary to help citizens lose weight, since cluster analysis showed that the MA cluster, which mainly monitors diet but lacks a daily and personalized calculation of EB, without feedback on controlling weight and anthropometric parameters, does not achieve significant weight loss. This study also established framework to define the optimal time and adherence to achieve clinically significant weight loss in a healthy population by increasing self-awareness of one’s own metabolic status, through the definition of a lower limit for adherence frequencies and the average time required to achieve significant weight loss. Overall, these data provide a framework for planning future digital-monitoring studies: accordingly, less stringent self-monitoring requirements for food tracking may positively affect adherence and weight loss. We also have evidence that, without explicit intervention (i.e., in providing exercise plans), physical activity, although a continuously monitored parameter, is not significantly altered.

This study has some advantages and limitations. This investigation used an open-access developed app with a professional database, which allowed for the objective collection of all participants’ self-monitoring data over the course of the study. All study components were remotely delivered, which facilitated the recruitment and retention of participants. However, limitations should also be noted. Although the population was very inhomogeneous in terms of age and starting BMI, recruitment was limited to Italian and Caucasian subjects working in the urban health system, thus limiting its generalizability. All data were subject to self-report bias, which we tried to limit through filtering and data-cleaning procedures, as described in the methods. Another problem may be represented by backlogging: digital food diaries are intended to be used in real-time, but diet recording can occur long after dietary consumption. Backlogging data can make diary entries appear as if they were recorded in real-time, when this may not have occurred. A significant correlation between percent BMI variations and recording dietary intake within 15 min of opening an electronic paper diary was found in a previous study [13]. Thus, it is possible that weight loss might have been greater if dietary monitoring occurred in a timelier manner. Although an examination of the association between the backlogging of data and weight change is outside the scope of the current study, a further investigation into the effect of backlogging self-monitoring data may be warranted. Lastly, the associations between the indicators of adherence with long-term weight loss should be investigated in large, randomized trials of a longer duration, and the integration of novel developed biomarkers of glucose and lipid metabolism [14,15,16,17,18,19] in the web app should be evaluated as a powerful improvement to studies of the effects and influence of dietary molecules.

## 5. Conclusions

Obesity and its metabolic complications are the most serious public health challenges of the 21st century. The prevalence of obesity has tripled in many countries. In the current pandemic, the issue of obesity has become more prominent, highlighting the need for its prevention. Finding and implementing solutions necessitates effective behaviour change. This study shows that integration of several IoT devices and a diet registry in a single app that can merge all the acquisitions into a single visualization dash, and a deeper analysis of digital behaviours, provides important information that can help one to correctly plan studies relying on digital applications. By relying on this information, a way of empowering citizens with scientific knowledge and validated instruments can be found, augmenting their self-awareness with the aim of achieving long-lasting results in the pursuit of a healthy lifestyle.

## Figures and Tables

**Figure 1 jpm-12-00568-f001:**
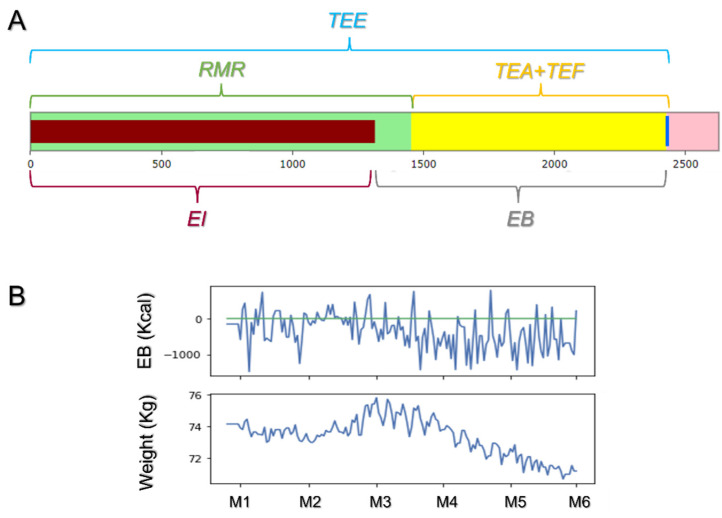
ArmOnIA control dashboard and EB/Weight time series. In the dashboard reported in (**A**), the green bar represents the estimated RMR, while the yellow bar constitutes the amount of additional energy obtained as the sum of TEF, deriving from energy expenditure related to food consumption, and TEA, that is, the energy consumed for other activities. To lose weight, the energy intake (overlapping red bar) must be kept under the vertical blue bar, which represents the value of the total energy expenditure (TEE). The difference between TEE and EI, indicated by the gray bracket, is the energy balance (EB), whose trend in time is reported in (**B**), together with weight records (below). These curves show the relationship between EB, expressed in Kcal, and weight maintenance, gain, or loss. If the EB is maintained around zero (green line), as between M1 and M2, no changes occur in the average weight. On the contrary, if the average EB remains above zero for a prolonged period (M2–M3), the weight increases, while an EB below zero (M3–M6) results in weight decrease.

**Figure 2 jpm-12-00568-f002:**
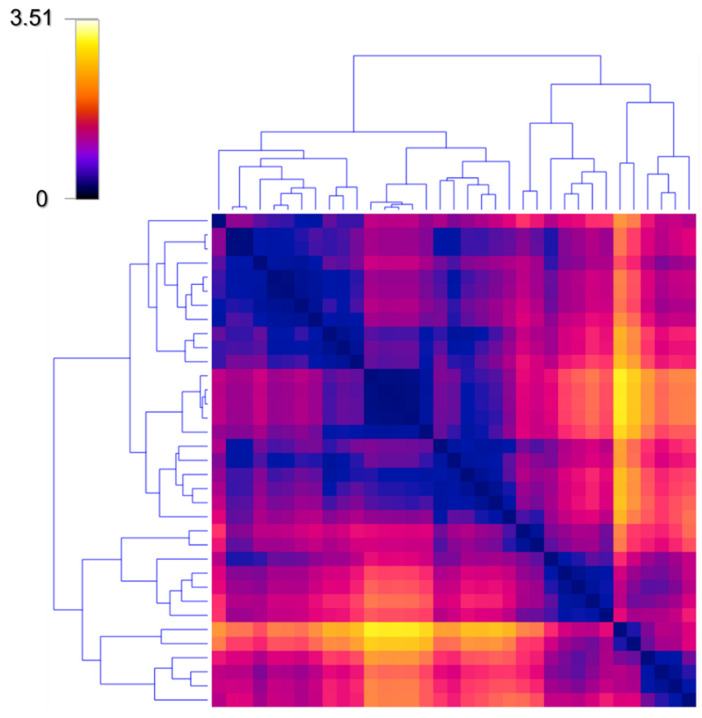
The distance map for hierarchical cluster analysis allows for distances between objects to be visualized using colored spots. In the map, the Euclidean distances between rows, corresponding to participants in the study, are represented in the weight- and food-frequency monitoring data, which constituted the features for clustering. According to the selected LUT (Blue–Magenta–Yellow), smaller distances are colored in blue (low values), while larger ones are shown in yellow (high values). The matrix is symmetric, and the diagonal is constituted by black spots, meaning that no user differs from himself in terms of WFM and FFM, respectively. In this representation, elements are arranged by clustering with ordered leaves, which maximizes the sum of similarities for adjacent elements.

**Figure 3 jpm-12-00568-f003:**
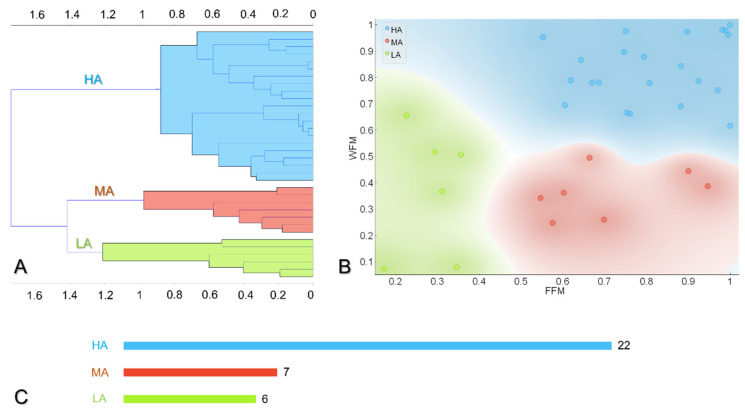
Unsupervised hierarchical clustering method and analysis. In (**A**), the hierarchical classification tree obtained through the Ward method is reported. The clustering identifies three different approaches, according to weight- and food-frequency monitoring values, as represented in (**B**). The first one, represented in light blue, is characterized by a high rate of weight- (0.83 ± 0.12) and food- (0.81 ± 0.15) monitoring, respectively, and is thus named high-adherence (HA); the second one is defined as medium-adherence (MA) and colored in red, showing a lower WFM (0.60 ± 0.09) and a higher FFM (0.70 ± 0.16); and the third one, the low-adherence (LA) group, in green, is characterized by the lowest self-monitoring frequencies, 0.37 ± 0.24 for weight, and 0.28 ± 0.07 for food, respectively. In (**C**) the number of study participants belonging to each cluster, indicated for HA (22/35, 63%), MA (7/35, 20%), and LA (6/35, 17%), respectively, reveals a good engagement of people in the self-monitoring study.

**Figure 4 jpm-12-00568-f004:**
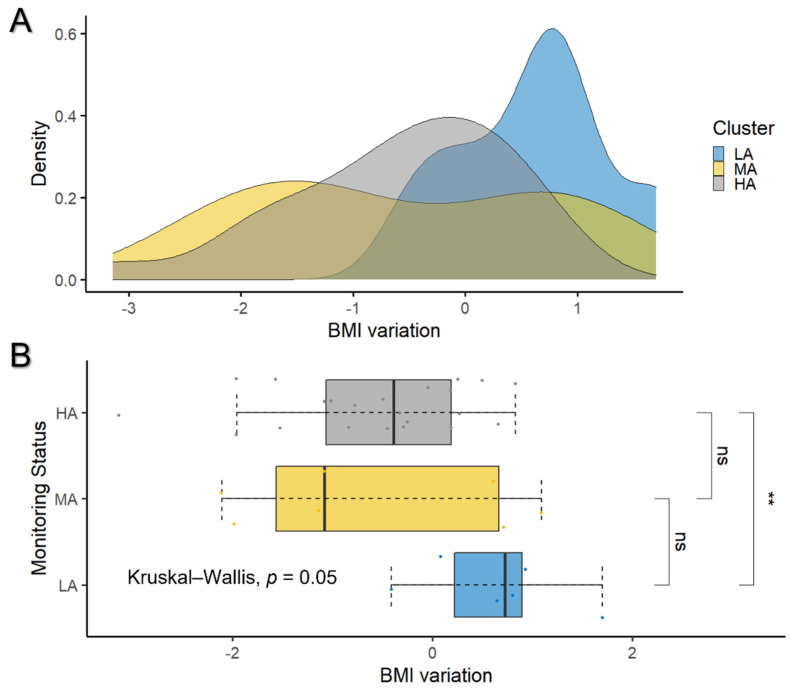
Distribution of BMI variation for different monitoring status. In (**A**) the density distribution (*y*-axis) of BMI variation (*x*-axis) is represented for HA (gray), MA (yellow), and LA cluster (blue), respectively. BMI values were standardized to µ = 0 and σ^2^ = 1. The corresponding box plot, together with the statistical analysis, is represented in (**B**). Kruskal–Wallis test, used to compare the three groups, shows a statistically significant (*p*-value = 0.05) difference in BMI variation among clusters. The result of the post-hoc comparison is then reported, revealing that a high-adherence rate of self-monitoring ensures a significant BMI decrease with respect to low-adherence rate (** stands for *p*-value = 0.0071).

**Figure 5 jpm-12-00568-f005:**
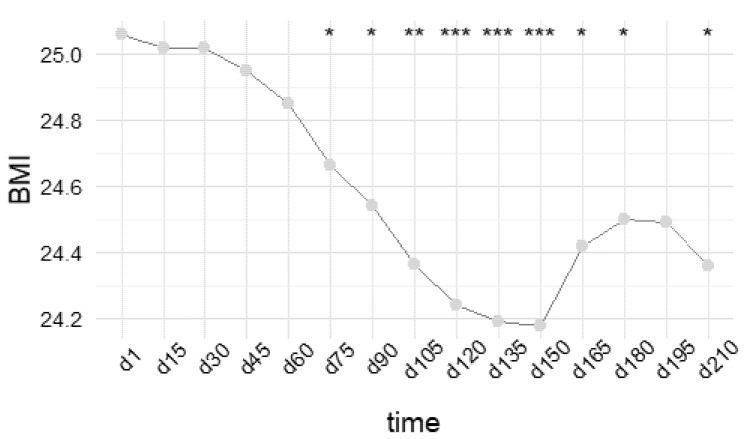
BMI variation in time allows the minimum average time of self-monitoring required to achieve a significant weight loss to be defined. The graph reports the mean BMI evaluated every 15 days (*y*-axis) as a function of time (*x*-axis) in a subgroup of the HA cluster. A repeated-measures ANOVA, followed by a paired *t*-test, revealed that after 75 days of high-rate self-monitoring, a significant BMI variation (*p*-value = 0.023) occurs, with mean values decreasing from 25.1 ± 3.2 (day 1) to 24.7 ± 3.4 (day 75). Interestingly, a constant monitoring longer than 75 days results in a continuous decrease until a minimum value is recorded after 150 days. At the end of the study period (210 days), the average BMI is still significantly lower (*p*-value = 0.01) with respect to the starting value. (* stands for *p*-value < 0.05; ** stands for *p*-value < 0.01; *** stands for *p*-value < 0.001).

**Table 1 jpm-12-00568-t001:** Descriptive characteristics of the clusters.

		Monitoring Status Group				Post-Hoc Comparison		
Variable	Overall (*n* = 35 ^1^)	LA (*n* = 6 ^1^)	MA (*n* = 7 ^1^)	HA (*n* = 22 ^1^)	*p*-Value ^2^ (Anova)	HA-LA	LA-MA	HA-MA
Sex					0.2			
Female	23/35 (66%)	2/6 (33%)	6/7 (86%)	15/22 (68%)				
Male	12/35 (34%)	4/6 (67%)	1/7 (14%)	7/22 (32%)				
Age	37 ± 12	37 ± 6	29 ± 6	39 ± 14	0.10			
Duration of the study [d]	160 ± 53	157 ± 58	140 ± 50	168 ± 54	0.3			
Average Daily Steps	8320 ± 2648	7172 ± 3379	8928 ± 2821	8440 ± 2418	0.4			
Starting BMI	24.6 ± 3.8	24.02 ± 3.1	23.7 ± 3.9	25.2 ± 4.0	0.5			
BMI variation	−0.4 ± 1.1	0.6 ± 0.7	−0.6 ± 1.3	−0.6 ± 0.9	**0.050**	**0.007 (**)**	0.07	0.96
Weight Frequency Monitoring (WFM)	0.66 ± 0.27	0.37 ± 0.24	0.6 ± 0.09	0.83 ± 0.12	**<0.001**	**0.004 (**)**	0.97	**<0.001 (***)**
Food Frequency Monitoring (FFM)	0.70 ± 0.24	0.28 ± 0.07	0.70 ± 0.16	0.81 ± 0.15	**<0.001**	**<0.001 (***)**	**<0.001 (***)**	0.14

^1^ Mean ± SD or Frequency (%); ^2^ Fisher’s exact test; Kruskal–Wallis rank sum test. Statistically significant differences are reported in bold. (**) stands for *p*-value < 0.01; (***) stands for *p*-value < 0.001.

## Data Availability

Not applicable.

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
