# Peer review of "Personalized Self-Monitoring of Energy Balance through Integration in a Web-Application of Dietary, Anthropometric, and Physical Activity Data"

_jpm, 2022, doi:10.3390/jpm12040568_

Round 1
Reviewer 1 Report
First I would like to congratulate the authors for such a current topic. Overall the article is easy to read without detecting major errors.
I think it would be interesting to review the discussion because in lines 300-302 and in lines 324 to 326 many presentations of numbers are made as a result and the aim is to discuss what was found, not to present data.
The last sentence in the conclusions should be taken out because nowhere in the work can you get information about what they write.
Congrats
Author Response
Response 1: We thank the reviewer for her/his positive comments and for the suggestions regarding the manuscript. We reviewed the discussion accordingly (see Results, Section 3.4, page 9, lines 283-288, and Discussion, page 10, lines 304-314 and lines 327-333).
Response 2: The last sentence in the conclusions has been moved to the Discussion section, since, as suggested by the reviewer, it provides a possible future development of the web app and not a Conclusion of the present work.

Reviewer 2 Report
The article deals with very important issue of obesity and its metabolic implications. Authors integrated several IoT devices and diet registry in a single app. It helps in deep analysis of digital behaviors and can empower citizens with proper scientific knowledge and validated instruments. It this way a healthy lifestyle can be achieved. The authors showed, that very important advancement could be the integration, in new app, of developed biomarkers of lipid metabolism to study the effects and the influence of diet. The paper is well written and English language is acceptable.
Very interesting paper, and it could be published in the present form after minor corrections.
Some important papers are missing and should be cited:
Patel ML, Wakayama LN, Bennett GG. Self-Monitoring via Digital Health in Weight Loss Interventions: A Systematic Review Among Adults with Overweight or Obesity. Obesity (Silver Spring). 2021 Mar;29(3):478-499. doi: 10.1002/oby.23088.
Harvey J, Krukowski R, Priest J, West D. Log Often, Lose More: Electronic Dietary Self-Monitoring for Weight Loss. Obesity (Silver Spring). 2019 Mar;27(3):380-384. doi: 10.1002/oby.22382.
Patel ML, Hopkins CM, Brooks TL, Bennett GG. Comparing Self-Monitoring Strategies for Weight Loss in a Smartphone App: Randomized Controlled Trial JMIR Mhealth Uhealth 2019;7(2):e12209, doi: 10.2196/12209
Author Response
Response 1: We thank the reviewer for her/his positive comments. As suggested, the indicated references have been added to the manuscript.
